# Dysregulated Coagulation in Parkinson’s Disease

**DOI:** 10.3390/cells13221874

**Published:** 2024-11-13

**Authors:** Xinqing Wang, Wenxin Li, Xinyue Zhao, Ning Hu, Xi Wang, Xilin Xiao, Kai Yang, Taolei Sun

**Affiliations:** 1School of Public Health, Hengyang Medical School, University of South China, Hengyang 421001, China; xinqing5@163.com; 2School of Chemistry, Chemical Engineering and Life Science, Wuhan University of Technology, 122 Luoshi Road, Wuhan 430070, China; 18747958960@163.com (W.L.); 291734@whut.edu.cn (X.Z.); 318526@whut.edu.cn (N.H.); 318544@whut.edu.cn (X.W.); suntl@whut.edu.cn (T.S.); 3State Key Laboratory of Chemo/Biosensing and Chemometrics, Hunan University, Changsha 410082, China; 4Hubei Key Laboratory of Nanomedicine for Neurodegenerative Diseases, School of Chemistry, Chemical Engineering and Life Science, Wuhan University of Technology, 122 Luoshi Road, Wuhan 430070, China; 5State Key Laboratory of Advanced Technology for Materials Synthesis and Processing, Wuhan University of Technology, 122 Luoshi Road, Wuhan 430070, China

**Keywords:** coagulation disorder, Parkinson’s disease, platelet, fibrinolysis, deep vein, thrombosis

## Abstract

Parkinson’s disease (PD), a prevalent neurodegenerative disorder characterized by dopaminergic neuron degeneration and α-synuclein accumulation, has been increasingly associated with coagulation dysfunction. This review synthesizes emerging evidence linking dysregulated coagulation to PD pathophysiology. We examine the alterations in coagulation parameters, including elevated fibrinogen levels, impaired fibrinolysis, and platelet dysfunction, which collectively contribute to a hypercoagulable state in PD patients. Epidemiological studies have revealed a higher incidence of thrombotic events, such as deep vein thrombosis (DVT) and stroke, among PD patients, suggesting significant comorbidity between PD and coagulation disorders. This review explores the potential pathophysiological mechanisms underlying this association, focusing on the roles of inflammation and oxidative stress. Additionally, we discuss the limitations of current research and propose future directions. This comprehensive analysis underscores the importance of understanding the coagulation–neurodegeneration axis in PD, which may lead to novel diagnostic and therapeutic strategies for this debilitating condition.

## 1. The Introduction

Parkinson’s disease (PD) ranks as the second most prevalent neurodegenerative disorder. Its characteristic features include the degeneration of dopaminergic neurons within the substantia nigra pars compacta (SNc) and the formation of Lewy bodies (LBs), intracytoplasmic aggregates containing misfolded α-synuclein (α-syn). The death of dopaminergic neurons within the SNc results in dopamine depletion in the striatum, contributing to the severe motor symptoms of PD, including bradykinesia, rigidity, and resting tremors [1]. Beyond motor impairments, PD patients also exhibit non-motor symptoms that significantly impact their quality of life, such as sleep disturbances, mood alterations, cognitive impairments, and autonomic dysfunction. These non-motor symptoms may manifest even before the onset of motor dysfunction [2]. PD is classified as one of the synucleinopathies, characterized by the central role of α-syn in its etiology. The occurrence of α-syn aggregation in PD patients implies disruption in α-syn proteostasis. Indeed, α-syn demonstrates a dynamic equilibrium in its multiple conformations, which is influenced by factors including oxidative stress, post-translational modifications, and lipids [3]. These factors tightly regulate the maintenance of the α-syn levels and its propensity for aggregation.

Currently, many drugs have been developed to treat PD. Among them, L-DOPA stands as a cornerstone therapy, aiming to restore the loss of dopamine and dopaminergic function. However, these drugs exhibit drawbacks, including diminishing efficacy over time, the emergence of medication-related complications such as motor fluctuations and dyskinesia (levodopa-induced dyskinesia, LID), and potential side effects like impulse control disorders and dopamine dysregulation syndrome [4]. Thus, innovative and more effective therapeutic approaches are required for PD treatment.

Coagulation processes and hemostasis are fundamental physiological mechanisms that ensure blood fluidity and prevent excessive bleeding upon vascular injury. Hemostasis encompasses a complex interplay of cellular and molecular events that culminate in the formation of a stable blood clot at the site of injury. This process involves a delicate balance between procoagulant and anticoagulant factors to maintain the blood flow while preventing thrombosis [5]. Three primary components contribute to hemostasis: the vascular endothelium, platelets, and coagulation factors. The intact vascular endothelium provides an antithrombotic surface and regulates the release of von Willebrand factor (vWF) and tissue factor (TF), crucial initiators of the coagulation cascade [6]. Platelets, circulating blood cells derived from megakaryocytes, play a pivotal role in primary hemostasis by adhering to the injured endothelium, forming a platelet plug, and releasing procoagulant factors [7,8]. Coagulation factors, synthesized primarily in the liver, orchestrate the conversion of soluble fibrinogen into insoluble fibrin, stabilizing the platelet plug and forming a mature blood clot [7]. Dysregulation of these processes can lead to thrombosis, contributing to various pathological conditions.

The coagulation cascade comprises a sequence of enzymatic reactions that culminate in the production of thrombin, a pivotal enzyme in clot formation. It involves two interconnected pathways: the intrinsic pathway, activated by contact with exposed collagen in injured blood vessels, and the extrinsic pathway, initiated by the release of TF from damaged endothelial cells (ECs). These pathways converge upon the activation of factor X, resulting in the conversion of prothrombin to thrombin. Thrombin serves as a central component in hemostasis, facilitating the conversion of fibrinogen to fibrin and activating platelets to augment clot formation. Thrombin generation is tightly regulated by endogenous anticoagulant mechanisms, such as antithrombin, protein C, and TF pathway inhibitor (TFPI), which restrict thrombin activity and prevent the formation of excessive clots [9]. Additionally, fibrinolysis, the enzymatic breakdown of fibrin clots catalyzed by plasmin, preserves the blood’s fluidity and facilitates clot removal upon the restoration of vascular integrity [10]. This complex interplay of procoagulant and anticoagulant factors ensures the preservation of the hemostatic equilibrium and vascular homeostasis.

Recent studies have increasingly pointed to the involvement of dysfunctional coagulation in the pathophysiology of PD. Both clinical observations and experimental models have provided compelling evidence that the coagulation cascade is dysregulated in PD [11,12,13], contributing to the disease’s complex pathology. A significant study by Ma et al. (2021) utilized pathway analysis to explore the differentially expressed proteins in two PD mouse models: one induced by the injection of α-syn preformed fibrils (PFF) and the other by the overexpression of the human A53T α-syn mutant. Both models are well-established representations of PD and recapitulate several key aspects of the disease, including protein aggregation and neuronal degeneration. The pathway analysis in this study revealed that the coagulation cascade is strongly implicated in PD pathogenesis [13]. In a parallel line of investigation, Infante et al. (2016) conducted an analysis of peripheral blood transcriptomes from patients with idiopathic PD as well as those with PD associated with the leucine-rich repeat kinase 2 (LRRK2) G2019S mutation, which is one of the most common genetic mutations linked to the disease. This study found a significant association between the coagulation cascade and PD across both patient groups. The involvement of the coagulation pathway in the transcriptomic profiles of PD patients supports the notion that systemic alterations in blood clotting mechanisms may be intricately linked to the pathogenesis of PD [12]. From a clinical perspective, Adams et al. (2019) provided direct evidence of altered coagulation in PD patients by examining specific parameters of blood clot formation. Their study demonstrated that blood samples from PD patients showed a significant increase in the initial rate of clot formation and a corresponding elevation in the alpha angle, which reflected the enhanced cross-linking of fibrin fibers. Moreover, the time required to reach the maximum rate of thrombus generation was reduced in these patients, indicating a hypercoagulable state [11].

This paper summarizes the evidence of dysregulated coagulation in PD, emphasizing changes in coagulation parameters including elevated fibrinogen, decreased fibrinolysis, and platelet dysfunction. The co-occurrence of coagulation disorders with PD, including deep venous thrombosis (DVT), stroke, and myocardial infarction (MI), is examined. Furthermore, potential pathophysiological mechanisms underlying this association are proposed. Moreover, the limitations of current research and further directions are also discussed.

## 2. Dysregulated Coagulation in PD

While traditionally viewed as a neurodegenerative disorder primarily affecting motor function, emerging evidence suggests that PD is associated with systemic alterations, including disruptions in the coagulation cascade [11]. Dysfunctional coagulation in PD represents a complex interplay between neurological, vascular, and inflammatory mechanisms [13], contributing to a heightened risk of thrombotic complications. Hypercoagulation in PD is evidenced by increased fibrinogen, impaired fibrinolysis, and platelet dysfunction.

### 2.1. Increased Fibrinogen

Elevated fibrinogen levels, a key factor in the coagulation cascade, have been implicated in the pathogenesis of PD (Table 1). Fibrinogen, as an essential component of clot formation, is thought to contribute to the increased risk of thrombus formation in PD, leading to a predisposition to vascular events.

Using the stable isotope labeling with amino acids in mammals (SILAM) technique, Ma et al. (2021) observed an increase in both fibrinogen α-chain isoform 1 precursor and fibrinogen β-chain precursor in α-syn PFF-injected mice and hA53T α-syn transgenic mice. These findings in animal models provide evidence that fibrinogen dysregulation may be a contributing factor in PD pathogenesis [13].

Epidemiological data have also provided substantial evidence supporting the link between elevated fibrinogen levels and PD. A pivotal study within the Honolulu Asia-Aging Study cohort identified a significant correlation between high fibrinogen levels and an increased risk of PD among elderly Japanese American men, particularly those aged over 75 years [14]. This study underscores the potential role of fibrinogen as a biomarker for PD susceptibility, particularly in aging populations.

In concordance with epidemiological findings, proteomic studies have consistently demonstrated elevated fibrinogen levels in both the plasma and cerebrospinal fluid (CSF) of PD patients. For instance, Lu et al. (2014) employed two-dimensional gel electrophoresis (2-DE) combined with liquid chromatography–tandem mass spectrometry (LC-MS/MS) to detect elevated concentrations of the fibrinogen γ-chain in the serum of PD patients [15]. Moreover, studies on peripheral blood lymphocytes isolated from PD patients revealed that different isoforms of the fibrinogen γ-chain correlate with the disease state and duration [16], suggesting a dynamic role of fibrinogen in PD progression. Additionally, an unbiased label-free LC-MS/MS analysis of CSF demonstrated an increase in both the fibrinogen β-chain and γ-chain levels in PD patients compared to controls [17], further supporting the involvement of fibrinogen in PD pathophysiology.

Morphological studies have also provided insights into the abnormal aggregation of fibrinogen in PD. Confocal microscopy studies have identified abnormal fibrinogen aggregation in the blood of PD patients [11], while scanning electron microscopy (SEM) revealed that fibrin clots in PD blood samples exhibited a distinct matted appearance, with the fibers being densely intertwined. Remarkably, this abnormal morphology could be reversed by the addition of lipopolysaccharide-binding protein (LBP), suggesting potential therapeutic interventions targeting fibrinogen aggregation [18]. Furthermore, postmortem examinations of the postcommissural putamen in PD patients revealed a significant 9.4-fold increase in the extravascular fibrinogen levels, indicating a possible contribution to neuroinflammation and neurodegeneration [19].

While many studies report elevated fibrinogen levels in PD, some investigations have yielded conflicting results, particularly regarding the fibrinogen β-chain in CSF. For example, two-dimensional difference gel electrophoresis (2D-DIGE) analysis revealed a decrease in both β-chain isoforms in the ventricular CSF of PD patients [20], and a similar reduction was observed in lumbar CSF using 2-DE [21]. In contrast, another study employing isobaric tagging for relative and absolute protein quantification (iTRAQ) coupled with MS/MS found no significant change in the fibrinogen β-chain in PD CSF [22]. These discrepancies could be attributed to alternative splicing and post-translational modifications of fibrinogen, which may affect its detection and function in different studies [25].

Beyond protein-level changes, biochemical studies have further implicated fibrinogen in PD. MicroRNAs (miRNAs), which regulate gene expression post-transcriptionally, have been shown to influence the fibrinogen levels in PD. Specifically, the downregulation of hsa-miR-144-3p in early PD patients suggests that the Fibrinogen Gamma Gene (FGG), a target of this miRNA, might be upregulated, potentially leading to increased fibrinogen levels [23].

Furthermore, post-translational modifications such as protein palmitoylation have been explored in the context of PD [24]. This modification, which involves the attachment of palmitic acid to cysteine residues, has been found to be upregulated in the fibrinogen polypeptides of the cerebral cortex in PD patients [24]. The increased palmitoylation of fibrinogen suggests alterations in its function or localization within the brain, which could contribute to the PD pathology. However, further studies are necessary to validate these findings and clarify the role of fibrinogen palmitoylation in PD.

### 2.2. Fibrinolysis Dysfunction

Dysfunction in fibrinolysis, the process by which blood clots are broken down, has been implicated as a contributing factor to the overall hemostatic imbalance observed in PD [26]. The pathological state of hypercoagulability is associated with an increase in plasminogen activator inhibitor-1 (PAI-1), a critical inhibitor of fibrinolysis [27]. PAI-1 exerts its inhibitory effects on both tissue plasminogen activator (tPA) and urokinase plasminogen activator (uPA), which are enzymes responsible for converting plasminogen into its active form, plasmin [28]. A decrease in plasmin activity subsequently leads to reduced fibrin degradation and an increase in clot formation, further exacerbating the hemostatic imbalance [29].

Although the role of fibrinolysis in PD has not been extensively explored, existing studies provide important insights. Notably, the PAI-1 levels have been observed to increase under inflammatory conditions and in the presence of extracellular α-syn aggregates [30,31]. In addition, plasmin can cleave extracellular α-syn, thereby reducing its aggregation and the formation of LB [30]. However, elevated PAI-1 levels in PD impede plasmin generation, leading to a reduction in α-syn proteolysis. This creates a pathological feedback loop where increased extracellular α-syn aggregation stimulates microglial and astrocytic inflammatory responses, which in turn elevate the PAI-1 levels and further reduce the plasmin activity. This cycle perpetuates α-syn aggregation and the progression of the PD pathology [32]. Further supporting the involvement of fibrinolytic dysfunction in PD, Lin et al. (2022) reported that treatment with α-syn PFFs in SH-SY5Y cells resulted in the decreased expression of tetranectin (TN) and increased expression of PAI-1, leading to reduced plasmin activity. This finding highlights the molecular interactions that contribute to impaired fibrinolysis in PD [33]. Additionally, clinical evidence from Sharma et al. (2021) demonstrated elevated levels of plasmin–antiplasmin complexes (PAP) in the plasma of PD patients, which is indicative of fibrinolytic system dysfunction in the disease [26].

### 2.3. Platelet Dysfunction

PD has been linked to several structural and functional changes in platelets, which may contribute to the disease’s complex pathophysiology (Table 2). The platelet count, a vital clinical parameter for the assessment of various diseases, is generally within the normal range in PD patients, with thrombocytopenia being a rare occurrence [34,35,36]. Studies by Giner et al. (2003) and Lee et al. (2013) have highlighted that thrombocytopenia in PD patients is often associated with long-term L-DOPA treatment, and the platelet counts tend to normalize after the cessation of L-DOPA [35,36], indicating a reversible drug-related effect.

Studies utilizing animal models have provided fundamental insights into platelet morphological alterations in PD. Investigations with α-synuclein knockout (α-syn −/−) mice have demonstrated significant platelet abnormalities, characterized by reduced platelet sizes, extensive degranulation, and notable fragmentation [37]. These preclinical observations have been complemented by substantial evidence from human studies. Kocer et al. (2013) documented increased platelet sizes in PD patients [38], while Factor et al. (1994) observed abnormally large platelet vacuoles, suggesting a potential mechanism underlying increased platelet volumes [39]. Subsequent clinical investigations have further corroborated these findings, demonstrating an elevated mean platelet volume (MPV) in PD patients compared to healthy controls [37,38]. These structural abnormalities may reflect the underlying platelet dysfunction in PD.

**Table 2 cells-13-01874-t002:** Platelet dysfunction in PD patients and transgenic mice.

Aspect of Platelet Function	Key Findings	Reference
Platelet Count	Generally normal in PD	[34]
	L-DOPA treatment associated with thrombocytopenia; effect is reversible upon treatment cessation	[35]
	L-DOPA treatment associated with thrombocytopenia; effect is reversible upon treatment cessation	[36]
Morphological Changes	Elevated mean platelet volume (MPV)	[38]
	Abnormally large platelet vacuoles	[37]
	Larger platelet size in PD patients; abnormally large platelet vacuoles	[39]
Platelet Adhesion	Defective adhesion observed; formation of defective platelet-rich thrombi	[40]
Platelet Activation	Hyperactivation contributes to increased blood coagulability	[11]
	Exogenous α-syn inhibits thrombin- or ionomycin-induced expression of P-selectin	[41]
Platelet Aggregation	Decreased aggregation responses to ADP and epinephrine	[42]
	MPP+ decreased platelet aggregation in platelet-rich plasma	[43]

Functionally, the platelets in PD patients exhibit significant alterations, particularly in adhesion, activation, and aggregation. Platelet adhesion, a critical step in thrombus formation, has been shown to be defective in α-syn −/− mice, leading to the formation of defective platelet-rich thrombi at sites of vascular injury [40]. This suggests that α-syn plays a role in maintaining normal platelet adhesion, and its absence or dysfunction could contribute to the coagulopathy observed in PD.

Platelet activation, a process driven by soluble agonists such as ADP, thromboxane A2 (TXA2), and thrombin, is another area of interest. Acquasaliente et al. (2022) found that exogenous α-syn inhibited the thrombin- or ionomycin-induced expression of P-selectin (CD62P), a key marker of platelet activation, on purified platelets [41]. This suggests that α-syn may modulate platelet activation pathways, and its dysregulation could exacerbate the hypercoagulable state in PD. Furthermore, Adams et al. (2019) demonstrated that platelet hyperactivation significantly contributes to the heightened blood coagulability seen in PD patients [11].

In terms of platelet aggregation, which is crucial for thrombus formation, PD patients exhibit decreased aggregation responses to ADP and epinephrine [42]. The inhibitory effects of exogenous α-syn on platelet aggregation, particularly through interference with the thrombin/proteinase-activated receptor 1 (PAR1) axis, further highlight the complex interplay between α-syn and platelet function [41]. Additionally, the neurotoxin MPP^+^, known to induce PD-like symptoms, has been shown to deplete the ATP levels, resulting in decreased platelet aggregation in platelet-rich plasma [43].

## 3. Comorbidity of Coagulation Disorders and PD

The comorbidity between coagulation disorders and PD has garnered significant attention due to its potential impact on the progression of PD and its clinical outcomes. Epidemiological studies have increasingly recognized the close association between PD and an elevated risk of coagulation abnormalities, such as DVT, stroke, and MI. Individuals with PD are more susceptible to these conditions compared to the general population (Table 3), suggesting an intrinsic link between neurodegenerative processes and vascular health.

DVT, characterized by thrombus formation in deep veins, presents a significant risk to patients with PD. These thrombi have the potential to dislodge and migrate to the pulmonary vasculature, resulting in pulmonary embolism (PE), a potentially life-threatening condition. Collectively, DVT and PE are classified as venous thromboembolism (VTE). It is hypothesized that early-stage PD patients, while not exhibiting balance disorders, manifest core symptoms such as bradykinesia, myotonia, and increased muscle tone [44], which may impair muscle pump function and venous blood return, potentially precipitating venous thrombosis [45]. As PD progresses, postural instability and prolonged immobility further exacerbate the risk of lower extremity venous thrombosis. The compromised muscle pump function and consequent blood flow stagnation significantly elevate the probability of thrombus formation and propagation [46].

Several studies have highlighted the heightened risk of DVT among PD patients [47,48,49,50]. For instance, Burbridge et al. (1999) identified asymptomatic DVT in 4.9% of a cohort of PD patients, a finding that has been reinforced by subsequent studies [48]. Yamane et al. (2013) conducted a cross-sectional study involving 114 asymptomatic PD outpatients and found DVT in 20.1% of these patients, with a bent knee posture identified as a significant risk factor [50]. Similarly, Nakajima et al. (2021) reported a 37.5% prevalence of DVT in a multicenter study of PD patients, underscoring the significance of this comorbidity [49]. Afsin et al. (2023) further investigated the direct relationship between PD and DVT, finding DVT and subsequent PE in 2.7% of patients after excluding those with other risk factors, suggesting that, while the prevalence may vary, the risk remains clinically relevant [47]. Additionally, Zibetti et al. (2010) documented asymptomatic DVT in PD patients shortly after undergoing stereotactic surgery for deep brain stimulation [51], emphasizing the importance of perioperative monitoring for thrombotic events. A recent study conducted by Li et al. (2024) examined the incidence of DVT in early-stage PD patients. The researchers reported that 9.4% of 117 PD patients were diagnosed with DVT, suggesting a potential correlation between PD progression and the DVT risk. This finding contributes to the growing body of evidence indicating an increased susceptibility to thrombotic events in the PD population, even in the early stages of the disease [52].

PE, in particular, has emerged as a critical cause of mortality in PD patients. Autopsy studies, such as those by Mosewich et al. (1994), have identified PE as the second most common cause of death in PD [53], highlighting the need for vigilance in recognizing and managing coagulation disorders in this population. Case reports have described instances where PE presented with acute psychosis, recurrent chest pain, and dyspnea in PD patients, further underscoring the clinical importance of early detection and intervention [54,55].

**Table 3 cells-13-01874-t003:** Comorbidity of coagulation disorders and PD.

Coagulation Disorder	Clinical Manifestations	Epidemiological Data	Reference
Deep Vein Thrombosis (DVT)			
		4.90%	[48]
		20%	[50]
	Blood clot formation in deep veins	37.50%	[49]
		2.70%	[47]
		4.90%	[51]
		9.40%	[52]
Pulmonary Embolism (PE)			
	Emboli block the arteries of the lung	23.10%	[53]
	Acute psychosis	N/A	[54]
	Chest pain and dyspnea	N/A	[55]
Stroke			
	Both ischemic and hemorrhagic presentations	Increased risk of stroke	[56]
	Stroke	Increased risk of stroke	[57]
	Ischemic stroke	Reduced stroke risk	[58]
	Stroke	No relationship	[59]
Myocardial Infarction (MI)			
	Reduced/blocked blood flow to heart	Reduced risk of MI	[60]

Stroke, another major comorbidity associated with PD, includes both ischemic and hemorrhagic subtypes. The relationship between stroke and PD is complex and has been the subject of numerous studies with varying conclusions. Autopsy studies frequently document the coexistence of stroke with the PD pathology. Some studies, such as those by Becker et al. (2010) and Skeie et al. (2013), suggest an increased risk of stroke following a PD diagnosis [56,57]. Conversely, others, like Struck et al. (1990), indicate a reduced stroke risk during the lifetime of PD patients [58], and some find no clear relationship between the two conditions [59].

MI, a condition resulting from reduced or blocked blood flow to the heart, appears to have a different relationship with PD. Some studies, such as that by Nabizadeh et al. (2023), suggest that PD patients may have a reduced risk of MI, possibly due to the lower prevalence of certain cardiac risk factors in this population [60]. However, the underlying mechanisms remain unclear and warrant further investigation.

Other coagulation disorders, although less commonly associated with PD, also present notable clinical challenges. Hemophilia, a rare bleeding disorder, has been observed in PD patients, with symptoms alleviated through deep brain stimulation and recombinant factor VIII infusion, as documented by Boehlen et al. (2017) [61]. Antiphospholipid syndrome (APS), characterized by the presence of antiphospholipid antibodies and associated with thrombotic events, has been reported in a few cases of PD, suggesting a potential link between the two conditions [62,63,64,65,66]. Additionally, disseminated intravascular coagulation (DIC), an acquired syndrome marked by widespread intravascular coagulation, has been reported in the context of neuroleptic malignant syndrome (NMS) during antiparkinsonian drug administration, illustrating the complex interplay between PD, its treatment, and coagulation disorders [67].

## 4. The Pathophysiological Mechanisms Underlying Dysfunctional Coagulation in PD

The exact mechanisms underlying the dysfunctional coagulation in PD remain unclear. However, researchers have proposed several hypotheses, including inflammation and oxidative stress.

### 4.1. Inflammation

The precise mechanisms underlying coagulation dysfunction in PD remain largely elusive; however, the prevailing consensus suggests a significant contribution from heightened inflammation (Figure 1). Both genetic and environmental factors associated with PD have been implicated in modulating immune function and inflammatory responses. Genetic mutations and post-translational modifications of α-syn can trigger conformational changes, leading to the formation of insoluble plaques [3]. These aggregates subsequently activate microglia and induce inflammatory responses. Notably, exposure to α-syn-activated microglia has been shown to prompt murine nigral dopaminergic neurons to increase major histocompatibility complex (MHC) class I expression, potentially marking them for destruction by antigen-specific CD8+ T cells [68]. Environmental factors, which play a significant role in PD’s etiology, also contribute to inflammation through various mechanisms. Exposure to certain environmental toxins and pollutants has been associated with an increased risk of PD development and progression, partly due to their ability to induce inflammatory responses in the brain [69]. For instance, pesticides such as rotenone and paraquat have been extensively studied for their neurotoxic effects, which include the induction of oxidative stress and inflammation in dopaminergic neurons [70].

Recent advancements in our understanding of inflammation have led to its recognition as both a trigger and sustainer of coagulation processes, extending beyond its traditional role as a response to tissue damage or infection. Inflammatory triggers induce the expression of TF on leukocytes, initiating the endothelial activation and recruitment of additional leukocytes [71]. Moreover, proinflammatory cytokines, such as interleukin-1β (IL-1β) and tumor necrosis factor-alpha (TNF-α), released in response to inflammation, activate endothelial cells (ECs) and facilitate leukocyte recruitment [72]. Endothelial dysfunction, exacerbated by inflammation, further enhances platelet activation and aggregation, thereby amplifying the coagulation cascade [73].

Recently, Laursen et al. (2024) reported significant findings in their study utilizing a mouse model combining lipopolysaccharide (LPS) treatment and α-syn PFF injection. Their results demonstrated the overrepresentation of fibrin clotting-related pathways in the context of inflammation in these LPS-treated, PFF-injected mice. Furthermore, a Western blot analysis revealed elevated levels of fibrinogen in the brain tissue of PFF-exposed subjects [74]. These observations suggest the potential intricate interplay between the α-syn pathology, neuroinflammation, and cerebral coagulation processes.

### 4.2. Oxidative Stress

Another possible mechanism through which PD contributes to coagulation dysfunction is via oxidative stress (Figure 1). The progressive loss of dopaminergic neurons in the SNc is accompanied by mitochondrial dysfunction and the increased production of reactive oxygen species (ROS) [1]. Elevated levels of oxidative stress markers, such as 8-hydroxy-2′-deoxyguanosine (8-OHdG) and malondialdehyde (MDA), have been consistently observed in PD patients, correlating with the disease’s severity and progression [75].

The impact of oxidative stress on coagulation is multifaceted. Firstly, the oxidative modification of coagulation factors can alter their structure and function, potentially leading to a hypercoagulable state. For instance, the oxidation of fibrinogen has been shown to enhance its thrombogenic potential by increasing its susceptibility to thrombin-induced cleavage and promoting the formation of denser and less permeable fibrin clots [76]. Secondly, oxidative stress can induce endothelial dysfunction. Endothelial cells exposed to high levels of ROS exhibit the reduced production of nitric oxide and increased expression of adhesion molecules, promoting platelet activation and aggregation [77]. Furthermore, oxidative stress-induced damage to platelets can enhance their reactivity and aggregation propensity, contributing to the overall prothrombotic state observed in PD [78].

## 5. The Limitations of Current Studies and Further Directions

The emerging field investigating coagulation dysfunction in PD pathophysiology offers promising insights but faces significant challenges. The current limitations include inconsistencies in the findings across studies, potentially stemming from variations in the experimental design and methodologies; the incomplete mechanistic understanding of coagulation abnormalities in PD and their contributions to disease progression; and difficulties in translating preclinical findings to clinical applications.

The intricate relationship between smoking, PD, and coagulation cascades presents a compelling paradigm in clinical neurology, characterized by both therapeutic potential and inherent complexities. Epidemiological investigations consistently demonstrate that smoking confers substantial neuroprotection, reducing the PD risk by 30–60% [79,80] through multiple mechanisms, including monoamine oxidase (MAO) inhibition, cytochrome P450 induction, and the nicotine-mediated preservation of dopaminergic neurons [81]. However, this protective effect exists in tension with smoking’s potential to modulate coagulation parameters through pro-thrombotic mechanisms and endothelial dysfunction. The complexity of this relationship is further evidenced by smoking’s influence on platelet function via MAO-mediated pathways [82,83], suggesting sophisticated interactions between neuroprotective mechanisms and coagulation cascades. These interactions necessitate rigorous individual risk stratification and potentially modified anticoagulation protocols for smoking PD patients. While the epidemiological evidence for smoking’s protective effect against PD is robust, the precise mechanisms by which it influences the coagulation parameters in PD patients remain incompletely characterized, highlighting the need for mechanistic studies to elucidate the molecular pathways linking smoking’s neuroprotective effects with coagulation cascades.

Despite these constraints, exploring the role of coagulation dysfunction in PD remains a critical avenue of research, potentially offering valuable insights for the development of novel diagnostic biomarkers and therapeutic strategies. Future research directions should focus on elucidating the molecular mechanisms linking coagulation dysfunction to PD progression, developing coagulation-based biomarkers for early diagnosis, and evaluating the potential of anticoagulant and antiplatelet therapies in PD treatment. This necessitates a multifaceted approach incorporating both preclinical and clinical investigations. Preclinical studies should utilize advanced animal models of PD, such as α-syn transgenic mice, to longitudinally profile the coagulation factor levels and activity across different disease stages. In vitro experiments examining the effects of various coagulation factors on α-syn aggregation and neuroinflammation in cultured neurons and microglia will provide crucial insights into the underlying molecular pathways.

Concurrently, large-scale prospective cohort studies should track changes in blood coagulation factors in healthy populations, coupled with neuroimaging techniques, to assess their predictive value for the PD risk and correlate them with early dopaminergic neuron loss. The relationship between coagulation dysfunction and non-motor PD symptoms represents another critical area for investigation. Cross-sectional studies in PD patients, analyzing the correlations between blood coagulation parameters and non-motor symptoms, complemented by functional neuroimaging, could reveal important associations. Furthermore, comparative analyses of the coagulation profiles in patients with PD, AD, and other neurodegenerative disorders may identify common and disease-specific alterations, potentially uncovering shared pathogenic mechanisms.

The development of novel therapeutic strategies targeting the coagulation–neurodegeneration axis should be actively pursued. This may include the design and screening of multifunctional molecules capable of modulating both coagulation and neuroprotective pathways, the development of targeted delivery systems to enhance the efficacy of anticoagulant drugs in the central nervous system, and the exploration of gene therapy or RNA interference techniques for the specific modulation of the coagulation-related gene expression in the PD brain. These innovative approaches have the potential to open new avenues for PD treatment that address both the neurodegenerative and vascular aspects of the disease.

## Figures and Tables

**Figure 1 cells-13-01874-f001:**
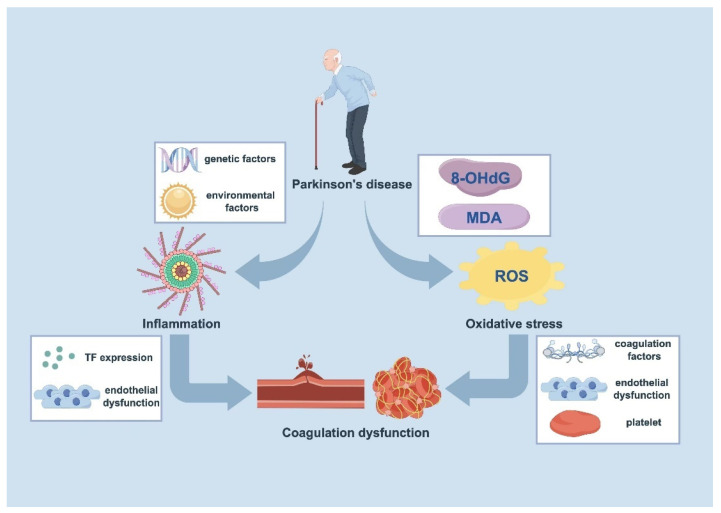
Inflammation and/or oxidative stress underpins the coagulation dysfunction observed in PD (By Figdraw). The complex etiology of PD, encompassing both genetic predisposition and environmental risk factors, initiates a cascade of pathophysiological events culminating in coagulation dysfunction. This process operates through two primary mechanisms: inflammatory dysregulation and oxidative stress. The pro-inflammatory state characteristic of PD induces TF expression and promotes endothelial dysfunction, thereby disrupting normal hemostatic mechanisms. Concurrent with these inflammatory changes, PD is characterized by elevated oxidative stress markers, particularly 8-OHdG and MDA, which serve as quantifiable indicators of oxidative damage. These elevated markers initiate a tripartite pathophysiological response: the direct oxidative modification of coagulation factors, the further compromise of endothelial integrity, and significant alterations in platelet function. The convergence of these inflammatory and oxidative pathways creates a self-perpetuating cycle that fundamentally alters coagulation homeostasis in PD patients.

**Table 1 cells-13-01874-t001:** Elevated fibrinogen in PD patients and transgenic mice.

	Samples	Detection Technique	Coagulation Parameters	Changes	Reference
PD transgenic mouse					
	Ventral midbrain	SILAM	fibrinogen α-chain isoform 1 precursor and fibrinogen β-chain precursor	increased	[13]
PD patients					
	Blood	BBL fibrometer	fibrinogen	increased	[14]
	Serum	2-DE combined with LC-MS/MS	fibrinogen γ-chain	increased	[15]
	Peripheral blood lymphocytes	2-DE combined with LC-MS/MS	fibrinogen γ-chain	increased	[16]
	CSF	Unbiased label-free LC-MS/MS	fibrinogen β-chain and γ-chain	increased	[17]
	Blood	Confocal microscopy	fibrinogen	abnormal fibrinogen aggregation	[11]
	Blood	Scanning electron microscopy	fibrinogen	exhibited a characteristic matted appearance	[18]
	Postcommissural putamen	Immunofluorescent microscopy	extravascular fibrinogen	increased	[19]
	Ventricular CSF	2D-DIGE	fibrinogen β-chain	decreased	[20]
	Lumbar CSF	2DE	fibrinogen β-chain	decreased	[21]
	CSF	iTRAQ in conjunction with multidimensional chromatography, followed by MS/MS	fibrinogen β-chain	no change	[22]
	Serum	miRNA sequencing	fibrinogen γ-gene	increased	[23]
	Cerebral cortex	Palmitome	palmitoylated fibrinogen	increased	[24]

2-DE: two-dimensional electrophoresis; LC-MS/MS: liquid chromatography–tandem mass spectrometry; iTRAQ: isobaric tags for relative and absolute quantitation; SILAM: stable isotope labeling by amino acids in mammals.

## Data Availability

No new data were created or analyzed in this study. Data sharing is not applicable to this article.

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
