# Peer review of "Dysregulated Coagulation in Parkinson’s Disease"

_cells, 2024, doi:10.3390/cells13221874_

Round 1

Reviewer 1 Report

Comments and Suggestions for Authors

Reviewer comments and suggestions

This review examines the association between Parkinson's disease (PD) and coagulation dysfunction, highlighting alterations in coagulation parameters such as elevated fibrinogen levels, impaired fibrinolysis, and platelet dysfunction. It further discusses the increased incidence of thrombotic events in PD patients, suggesting a significant comorbidity between PD and coagulation disorders.

Recommendation

1.      The objective is clearly defined and relevant to the current context.

2.      The article is well-written and effectively discussed with previously published papers.

Comments and suggestions

3.      The article does not adequately address the therapeutic role. If published data are available demonstrating its potential as a therapeutic target, these should be incorporated into the revised manuscript.

4.      Provide suitable citation for lines 125-128.

5.      Provide suitable citation for lines 180-183.

6.      Provide suitable reference/s for lines 192-199.

7.      Figures 1 and 2 do not add valuable information and appear ineffective. Both figures require a revision.

8.      Check reference style and do the needful correction.

Author Response

Response to Reviewer 1:

1. This review examines the association between Parkinson's disease (PD) and coagulation dysfunction, highlighting alterations in coagulation parameters such as elevated fibrinogen levels, impaired fibrinolysis, and platelet dysfunction. It further discusses the increased incidence of thrombotic events in PD patients, suggesting a significant comorbidity between PD and coagulation disorders.

Recommendation

  1. The objective is clearly defined and relevant to the current context.
  2. The article is well-written and effectively discussed with previously published papers.

We thank the reviewer for these positive comments.

2. The article does not adequately address the therapeutic role. If published data are available demonstrating its potential as a therapeutic target, these should be incorporated into the revised manuscript.

Thank you for this valuable suggestion. Although no published data currently explore its potential as a therapeutic target, we have addressed its therapeutic potential in the "Future Directions" section. Specifically, we propose the development of novel therapeutic strategies targeting the coagulation-neurodegeneration axis (See page 12, line 449-454), including the following approaches:

-Designing and screening multifunctional molecules that can simultaneously modulate coagulation and neuroprotective pathways.

-Developing targeted delivery systems to enhance the efficacy of anticoagulant drugs within the central nervous system.

-Exploring gene therapy or RNA interference techniques to selectively modulate the expression of coagulation-related genes in the PD brain.

3. Provide suitable citation for lines 125-128.

 We have added appropriate citations to support our statements about PD is associated with dysfunctional coagulation. See page 3, line 125-129.

4. Provide suitable citation for lines 180-183.

 We have added relevant citations to support our statements about the role of palmitoylation in PD. See page 5, line 189-192.

5. Provide suitable reference/s for lines 192-199.

 We have added appropriate citations to support our discussion about fibrinolysis. See page 6, line 197-205.

6. Figures 1 and 2 do not add valuable information and appear ineffective. Both figures require a revision.

 Thank you for the suggestion. We have combined the two figures into one and substantially enhanced it with more detailed visual elements and clearer pathway representations. See page 10, figure 1.

7. Check reference style and do the needful correction.

 We have carefully reviewed and corrected all references to ensure they follow the journal's formatting guidelines. See references.

Reviewer 2 Report

Comments and Suggestions for Authors

I read with interest this review exploring the link between Parkinson's disease (PD) and coagulation dysfunction. The authors first present evidence of alterations in coagulation parameters in PD, including elevated fibrinogen levels, reduced fibrinolysis, and platelet dysfunction. They then examine the co-occurrence of coagulation-related disorders with PD. Furthermore, the review proposes potential pathophysiological mechanisms that may underlie this association. Finally, the authors address the limitations of current research and suggest directions for future investigation.

Overall, the paper is well-written, with a clear structure and logical flow. The rationale for the study is effectively presented, and the manuscript thoughtfully highlights both the strengths and limitations of the existing research on this topic. I have a few suggestions that could further enhance the quality of the paper before publication.

-        In each section of the manuscript, consider presenting findings from animal models of PD first, followed by results from studies on human PD patients. This organization may enhance the logical progression of the evidence.

-        Given that smoking is known to have a protective effect against PD (see for example 10.1016/0304-3940(91)90804-3; 10.1016/j.parkreldis.2024.107022; 10.1002/mds.20117), it would be valuable to discuss its role in relation to coagulation-related disorders in PD patients. Exploring this aspect could add an interesting dimension to the paper.

-        The manuscript would benefit from adding Tables similar to Table 1. For instance, one table could focus on factors other than elevated fibrinogen, while another could cover PD-related comorbidities of coagulation disorders. Additionally, please include a legend for Table 1 that defines all abbreviations used.

-        The current figures lack sufficient detail. A potential improvement could be to combine the two figures into one, incorporating more relevant information—perhaps replacing text with visual elements for clarity.

-        On page 3, in the paragraph title, please capitalize the term ‘Increased’. Similarly, ensure the word ‘Coagulation’ in Table 1 is capitalized for consistency.

Author Response

Response to Reviewer 2

1. I read with interest this review exploring the link between Parkinson's disease (PD) and coagulation dysfunction. The authors first present evidence of alterations in coagulation parameters in PD, including elevated fibrinogen levels, reduced fibrinolysis, and platelet dysfunction. They then examine the co-occurrence of coagulation-related disorders with PD. Furthermore, the review proposes potential pathophysiological mechanisms that may underlie this association. Finally, the authors address the limitations of current research and suggest directions for future investigation.

 We appreciate these positive comments

2. Overall, the paper is well-written, with a clear structure and logical flow. The rationale for the study is effectively presented, and the manuscript thoughtfully highlights both the strengths and limitations of the existing research on this topic. I have a few suggestions that could further enhance the quality of the paper before publication.

In each section of the manuscript, consider presenting findings from animal models of PD first, followed by results from studies on human PD patients. This organization may enhance the logical progression of the evidence.

 We have reorganized the presentation of evidence throughout the manuscript to follow this logical progression from animal models to human studies, particularly in sections discussing fibrinogen levels, fibrinolysis, and platelet dysfunction. See page 3, line 132 to page 8, line 264.

3. Given that smoking is known to have a protective effect against PD (see for example 10.1016/0304-3940(91)90804-3; 10.1016/j.parkreldis.2024.107022; 10.1002/mds.20117), it would be valuable to discuss its role in relation to coagulation-related disorders in PD patients. Exploring this aspect could add an interesting dimension to the paper.

 This point is well taken. Now we add one paragraph in the section “The limitation of current studies and Further direction” to discuss the intricate relationship between smoking, PD, and coagulation cascades. See page 11, line 409 to page 12, line 425.

4. The manuscript would benefit from adding Tables similar to Table 1. For instance, one table could focus on factors other than elevated fibrinogen, while another could cover PD-related comorbidities of coagulation disorders. Additionally, please include a legend for Table 1 that defines all abbreviations used.

 Thanks for the reviewer’s suggests. We have:

  • Added table 2 summarizing platelet dysfunctions in PD. See page 7, table 2.
  • Created table 3 summarizing PD-related coagulation disorder comorbidities. See page 9, table 3.
  • Added a comprehensive legend to Table 1 with all abbreviations defined. See page 4, table 1.

5. The current figures lack sufficient detail. A potential improvement could be to combine the two figures into one, incorporating more relevant information—perhaps replacing text with visual elements for clarity.

 See the response to reviewer 1, comment 6.

6. On page 3, in the paragraph title, please capitalize the term ‘Increased’. Similarly, ensure the word ‘Coagulation’ in Table 1 is capitalized for consistency.

 Thanks for your comments, we have made these capitalization corrections for consistency throughout the manuscript. See page 4, table 1.